# Research on AZ80 + 0.4%Ce (wt %) Ultra-Thin-Walled Tubes of Magnesium Alloys: The Forming Process, Microstructure Evolution and Mechanical Properties

**Zhaoming Yan [1] , Min Fang [1],*, Zhendong Lian [1], Zhimin Zhang [1],*, Jiaxuan Zhu [2], Guanshi Zhang [1] and Yiding Wang [3]**

[1]   School of Material Science and Engineering, North University of China, Taiyuan 030051, China; zmyan1027@126.com (Z.Y.); xiaozuozhen7985@163.com (Z.L.); s1403035@st.nuc.edu.cn (G.Z.)
[2]   College of Mechatronics Engineering, North University of China, Taiyuan 030051, China; zhujiaxuan66@163.com
[3]   State Key Laboratory of Solidification Processing, Northwestern Polytechnical University, Xi'an 710072, China; npu_ydwang@163.com
*   Correspondence: minfang_nuc@126.com (M.F.); zhangzhimin@nuc.edu.cn (Z.Z.); Tel.: +86-151-0346-8697 (M.F.)

**Abstract:** Ultra-thin-walled tubes of magnesium alloys have received more and more attention in producing precision components for medical devices. Therefore, thin-walled tubes with high quality are desperately needed. In this study, the process of multi-pass variable wall thickness extrusion was carried out on an AZ80 + 0.4%Ce Mg alloy with up to five passes—one-pass backward extrusion and four-pass extension—to fabricate the seamless thin-walled tube with an inside diameter of 6.0 mm and a wall thickness of 0.6 mm. The average grain size decreased from 46.3 μm to 8.9 μm at the appropriate deformation temperature of 350 °C with the punch speed of 0.1 mm/s. X-ray diffraction (XRD), optical microscope (OM), scanning electron microscopy (SEM), and the Vickers hardness (HV) tester were utilized to study the phases, microstructure, and hardness evolution. It can be observed that low deformation temperatures (240 °C and 270 °C) and low strain (1 pass extrusion and 1 pass extension) lead to twins that occupy the grains to coordinate deformation, and a slip system was activated with the accumulation of strain. The results of the Vickers hardness test showed that twinning, precipitation of second phases, twinning dynamic recrystallization (TDRX), and work hardening were combined to change the hardness of tubes at 240 °C and 270 °C. The hardness reached 93 HV after the third pass extension without annealing at 350 °C.

**Keywords:** ultra-thin-walled magnesium alloy tubes; multi-pass variable wall thickness extrusion; twinning; DRX; hardness

## 1. Introduction

With the advent of Industry 4.0, precision components are playing an important role in the field of electronic products and medical manufacturing. A tube is a common structure, while an ultra-thin walled tube is difficult to form directly. Meanwhile, the application of light-weight and high-strength materials is a key strategy to help solve the energy crisis [1–4].

Magnesium and magnesium alloys have become hot research topics because of their low density, elastic modulus matching with the particular structure, degradable absorption, and excellent biocompatibility [5–7]. Extrusion and cold drawing are the most widely used processes for tube production at present. However, due to the poor ductility of magnesium alloys at room temperature, the problems of a low finished product yield and the high cost of cold drawing tubes urgently need

to be addressed. Hot plastic deformation is a common method that is beneficial to the metal flow of magnesium alloys, as well as refinement of the microstructure by the accumulation of strain, corresponding to the characteristics of uniform deformation and excellent mechanical properties [8–10]. Because ultra-thin walled tubes usually have a thickness of 0.2–1.0 mm, they require high uniformity and are difficult to fabricate, especially the hexagonal closed-packed structure of the Mg alloy, which could increase the difficulty of manufacturing. A few attempts and clinical applications have been made for the new technology. In previous studies, Wang et al. [11] successfully developed the hot indirect extrusion and multi-pass cold drawing methods to produce an outside diameter of 2.9 mm and a wall thickness of 0.2 mm with a relatively uniform microstructure. In the study of Hanada et al. [12], cold drawing was used to create Mg-0.8%Ca and AZ61 alloys thin-walled tubes, and stent with a 1.5–1.8 mm outer diameter and 150 μm thickness were fabricated. However, some studies showed that cold drawing can influence the rate of final products because of the bad plastic deformation ability of magnesium alloy at room temperature [13,14]. Other processes like equal-channel angular pressing (ECAP) were used in the investigation of Ge et al. [15]: sub-micrometer grain sized billets of ZM21 alloy with a yield strength of 340 MPa were obtained by two passes of ECAP and low-temperature extrusion, and a tube with an outer diameter of 4 mm and a thickness of 1 mm could be used in a biodegradable stent. Faraji et al. [16] succeeded in producing a 500 nm ultrafine-grained AZ91 magnesium alloy tube through severe plastic deformation by means of tubular equal channel angular pressing (TECAP). Based on the hexagonal closed-packed HCP structure of the Mg alloy, there are only {0001}<11$\bar{2}$0> slip systems that can move on the basal plane at room temperature, two of which are independent slip systems, while more than five independent slip systems are needed for the deformation to prevent cracks according to the Von Mises criterion [17–19]. Therefore, hot extrusion is more suitable than cold drawing, and the yield of final products is higher.

In the present work, a new process of hot multi-pass variable wall thickness extrusion is proposed to replace cold drawing to fabricate ultra-thin walled tubes. To demonstrate the applicability of the method, the AZ80 + 0.4Ce% magnesium alloy was chosen to execute by five passes. The microstructure evolution and hardness examination were investigated to verify the possibility of new processes and the mechanism of the deformation of tubes during the multi-pass variable wall thickness hot extrusion process.

## 2. Materials and Methods

### 2.1. Strengthening Mechanism

For increasing the strength and corrosion of magnesium alloys, grain refinement and precipitation strengthening, two important strengthening techniques during the deformation of magnesium alloy, are usually used. Based on the characteristics of the low stacking faults (SFs) of magnesium alloys, dynamic recrystallization (DRX) generally takes place during hot deformation [20,21]. As an important dynamic softening and grain refinement mechanism, DRX plays an important role in controlling the deformation structure, strengthening the plastic forming ability, and improving the mechanical properties of magnesium alloys [22]. Non-distorted small grains formed by DRX can improve the strength by the fine grain strengthening mechanism. Dispersion strengthening usually happens after the second-phase precipitate in the boundaries in the process of deformation. Pinning of the dispersed second phases can effectively strengthen the grain boundaries and hinder the movement of dislocation, which can improve the yield strength of alloys [23].

### 2.2. Materials and Molds

In this study, AZ80 + 0.4%Ce (wt %) Mg alloy (Yinguang Magnesium Industry Group Co., LTD, Yuncheng, China) was selected for the experimental investigation; the chemical compositions are shown in Table 1. Cylindrical specimens 10 mm in diameter and 10 mm in length were machined from the bar, which was pre-extruded beforehand.

**Table 1.** Chemical composition of AZ80 + 0.4%Ce magnesium alloy (wt %).

| Al | Zn | Ce | Mn | Si | Cu | Fe | Ni | Mg |
|------|------|------|------|--------|--------|---------|---------|---------|
| 8.90 | 0.53 | 0.40 | 0.20 | ≦0.01 | ≦0.01 | ≦0.005 | ≦0.005 | balance |

The dies were manufactured from 2CrMo8V steel and achieved 45 HRC after heat treatment. In order to keep the fine machinability of Mg alloy, a holding furnace was used throughout the process. Figure 1 shows the experiment molds for fabricating the thin-walled tube of Mg alloy. It can be seen that the quick disassembly structure was adopted to the molds, and the bottom die and variable diameter extension rings, which had four different sizes, were put in the fixed table to fabricate a stent with a thickness of 0.6 mm and an inner diameter of 6.0 mm. One of the characteristics of the forming process was that the workpiece could be stuck to the punch ready for the subsequent extending deformation without alignment. A key point that should be considered during the forming of thin-walled tubes is the uniformity of the wall thickness. The locating ring was designed in the molds, as shown in Figure 1a, to maintain uniformity. Another special structure in the molds was extension rings; a cooperative convex structure was designed to simplify the deformation process and accumulate area reduction, and the stacking structure is defined as follows [13]:

$$\psi\% = \frac{F_q - F_h}{F_q}\%,$$

in which the accumulation of area reduction is represented by $\psi\%$; the section area before and after deformation are presented by $F_q$ and $F_h$, respectively. Thus, the parameters in the formula are related to the reduction in wall thickness and the reduction in diameter. From Figure 2b, it is clear that the first pass of deformation was backward extrusion to form the bottle-shaped workpiece with a punch diameter of 6.0 mm and a bottom die diameter of 10.0 mm, and the following process of extension was accomplished by four extension rings with diameters of 9.0 mm, 8.4 mm, 7.8 mm, and 7.2 mm, respectively.

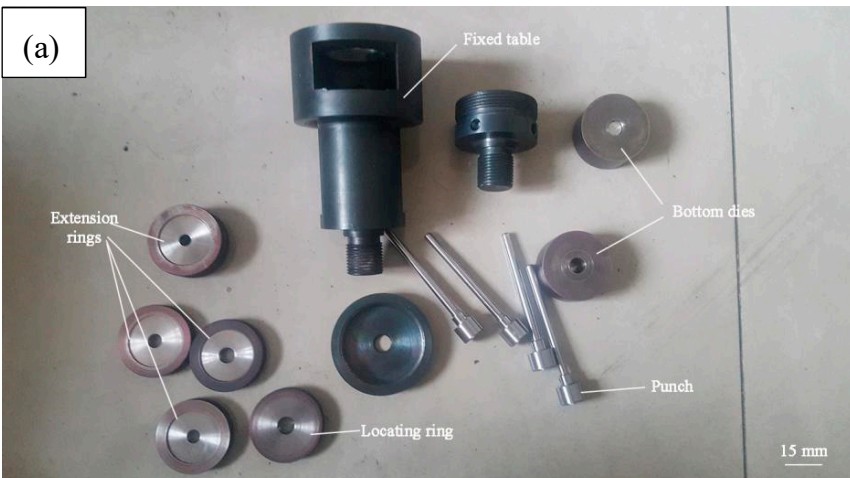

**Figure 1.** *Cont.*

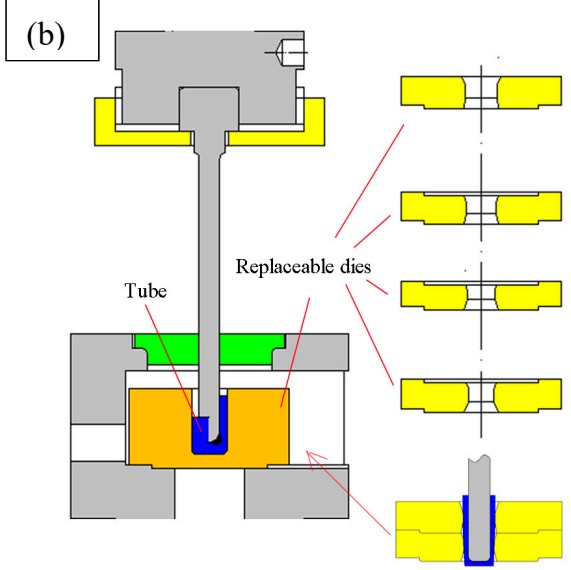

**Figure 1.** Experiment mold for fabricating thin-walled tubes: (**a**) die parts; (**b**) schematic diagram.

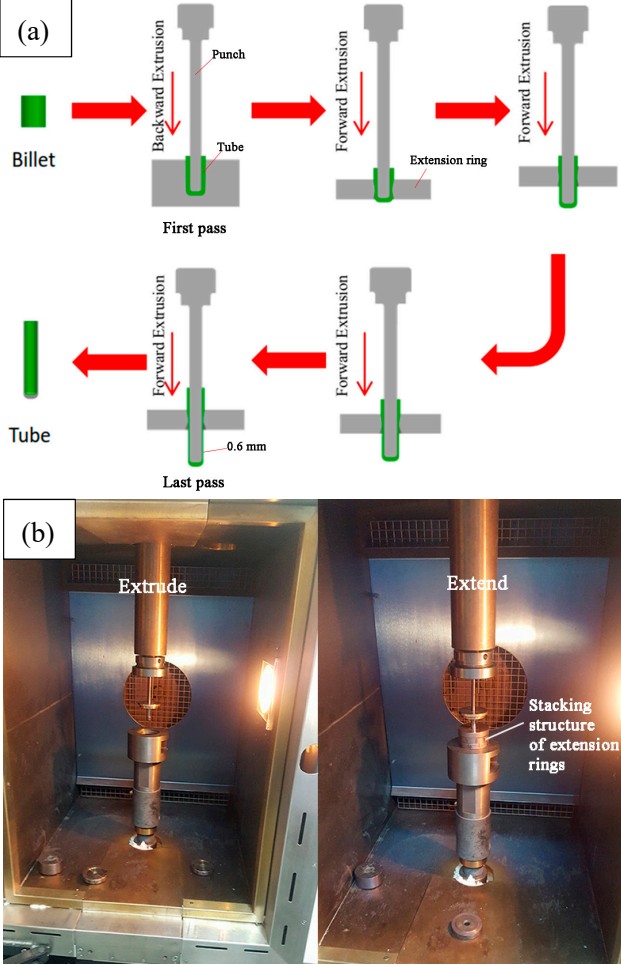

**Figure 2.** Forming process of magnesium alloy thin-walled tubes: (**a**) extension, one by one after backward extrusion; (**b**) stacking structure of extension rings after backward extrusion.

### 2.3. Experimental Procedures

Figure 2 shows the forming process of a thin-walled tube from a billet to a tube with a thickness of 0.6 mm. Five different temperatures ranging from 240 °C to 390 °C were chosen to conduct experiments with a 0.1 mm/s moving speed of the punch; billets and molds were heated at the same temperature, while the holding time varied a lot. The billets were usually held for 10 min and the molds for 1 h. After that, graphite lubricant was daubed on the punch and bottom die. Firstly, backward extrusion was performed and a workpiece with an inner diameter of 6.0 mm and a thickness of 2.0 mm was obtained directly. Then, forward extensions with area reduction percentages of 43.8%, 64.0%, 79.8%, and 91.0% were conducted one by one as in the process shown in Figure 2a; or as a stacking structure of extension rings, as shown in Figure 1b. Table 2 shows the cumulative area reduction percentage and the reduction of wall thickness; it can be seen that the wall thickness has been nearly halved from the beginning (2.0 mm) to the end (1.2 mm). After deformation, the tube was immediately quenched in deionized water at room temperature.

**Table 2.** Reduction of thickness and area reduction percentage during deformation.

| Pass | Inner Diameter/mm | Outer Diameter/mm | Wall Thickness Reduction/mm | Cumulative Area Reduction Percentage/% |
|------|------|------|------|------|
| 0 | 6.0 | 10.0 | | |
| 1 | 6.0 | 9.0 | 0.5 | 43.8 |
| 2 | 6.0 | 8.4 | 0.3 | 64.0 |
| 3 | 6.0 | 7.8 | 0.3 | 79.8 |
| 4 | 6.0 | 7.2 | 0.3 | 91.0 |

The thin-walled tubes after backward extrusion and four passes of extension are shown in Figure 3, and the change of wall thickness after each extension is shown in Figure 4. It can be seen that tubes with uniform thickness of 0.6 mm were manufactured after four passes extension. Meanwhile, the tolerance of wall thickness becomes smaller and smaller. To be specific, the machinability and ductility of AZ80+0.4%Ce increased with the accumulation of strain, corresponding to slide of non-basal slip and the weakening of texture [24].

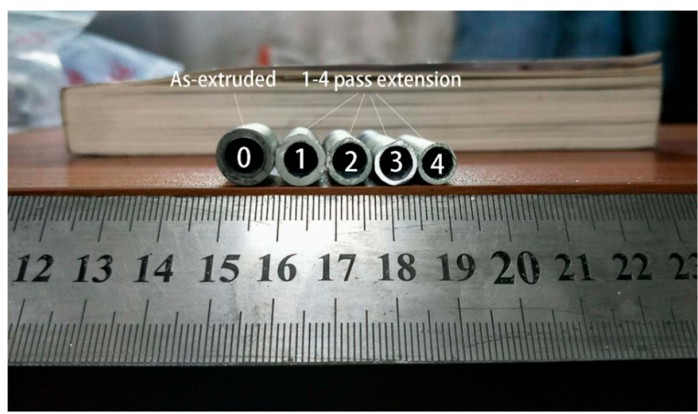

**Figure 3.** The extruded and four pass extended tubes.

Before observation, the metallographic samples were ground, polished, and chemically corroded in a solution of 0.5 g oxalate, 2.5 mL 99% acetic acid, and 60 mL deionized water. Then, the phase compositions were measured by an X-ray diffractometer (XRD, DX-2700, Fangyuan Inc., Dandong, China) with a diffraction angle ranging from 20° to 80° and a scanning speed of 10°/min. The microstructure was observed through an optical microscope (OM, A2m, Zeiss, Oberkochen, Germany) with a magnification range of 50–1000. The morphology and distribution of phases were investigated by

scanning electron microscopy (SEM, SU5000, Hitachi, Tokyo, Japan) at 20 Kv, and an energy-dispersive X-ray spectrometer (EDS, Genesis, EDAX Inc., Mahwah, NJ, USA) installed on SEM was operated to understand the atomic ratio of phases with a voltage of 20 Kv, working distance of 10 mm, and the element card of Mg, Al, Mn, and Ce.

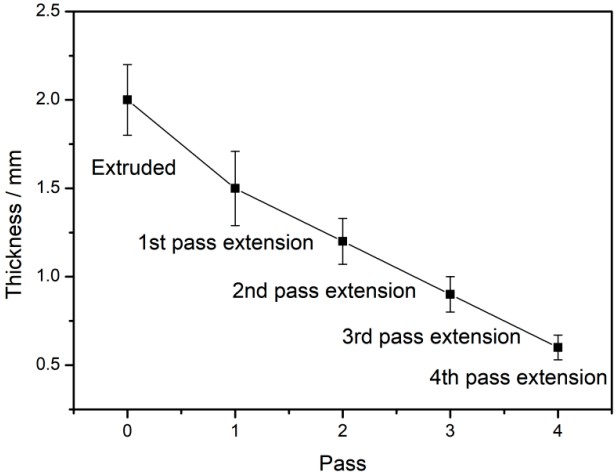

**Figure 4.** Change of wall thickness.

## 3. Results and Discussion

### 3.1. Microstructure Evolution

#### 3.1.1. Φ10 mm × 10 mm Original Extruded Bar

The microstructure of the as-extruded AZ80 + 0.4%Ce magnesium alloy is depicted in Figure 5. The average grain size is 46.3 μm before deformation, calculated by the linear intercept method. A large number of second phases gather at the boundaries; also, a small number of bulk-shaped and rod-shaped phases with high light are distributed irregularly in the matrix, as shown in Figure 5b. With the magnification increasing to 500× and then 1200×, it can be noticed that the strip-shaped and net-shaped second phases extend from the grain boundary to intragranular region.

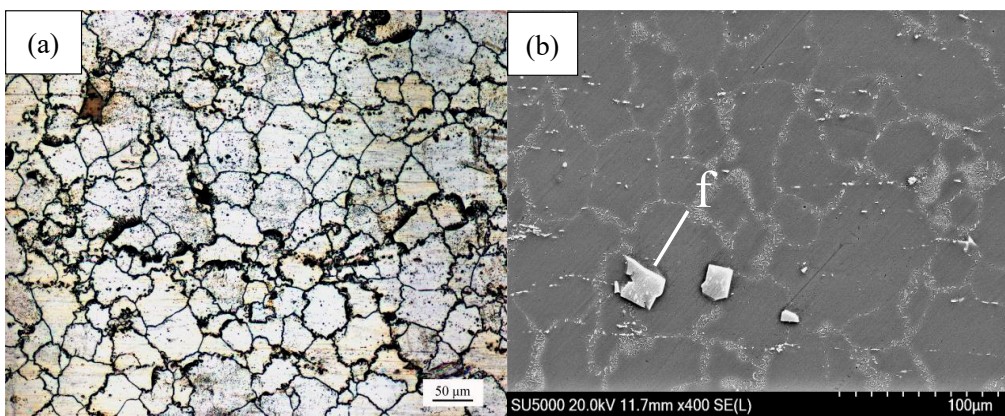

**Figure 5.** *Cont.*

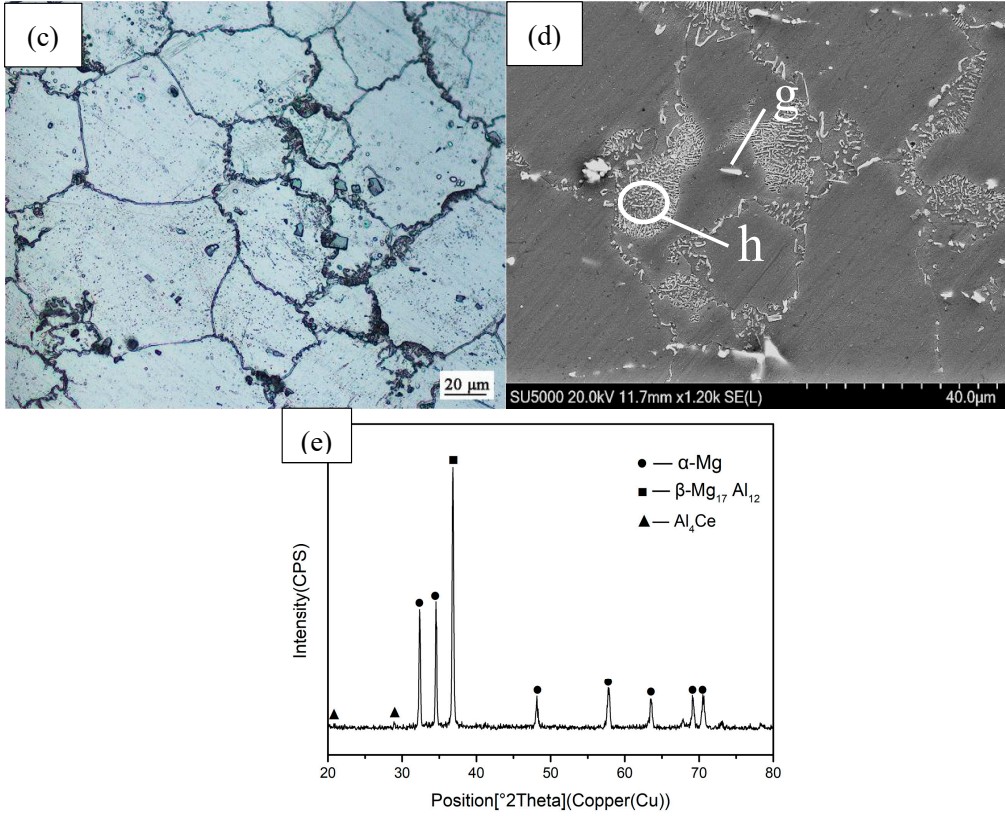

**Figure 5.** Microstructure of original extruded bar observed by OM and SEM: (**a**) OM with 200× magnification, (**b**) OM with 500× magnification, (**c**) SEM with 400× magnification, (**d**) SEM with 1200× magnification, (**e**) XRD pattern of the original extruded alloy.

Figure 5e shows the XRD pattern of the original pre-extruded alloy, and the EDS results with element atomic percentage of marked points in Figure 5b,d are shown in Table 3. It can be seen that the strip-shaped and net-shaped second phases that gather in the boundaries are β-$Mg_{17}Al_{12}$ (point h) and have the effect of hindering the growth of boundaries. The bulk-shaped phases marked with point f in the grains are $Al_8Mn_5$, which has too low a content in the matrix to be defined in the XRD pattern, while it can be defined in EDS. Another important phase is $Al_4Ce$ (point g), with the shape of a short rod or needle, whose formation is due to the addition of Ce [25].

**Table 3.** EDS results with the element atomic percentage of points in Figure 6b,d (at %).

| Points | Mg | Al | Mn | Ce |
|--------|-------|-------|-------|------|
| f | 8.34 | 45.34 | 36.44 | 9.88 |
| g | 47.97 | 40.94 | 9.35 | 0.15 |
| h | 55.24 | 34.99 | 9.69 | 0.08 |

### 3.1.2. Microstructure Evolution at the Same Temperature with Different Passes

Figure 6 depicts the microstructure evolution with the deformation of extrusion and four-pass extension at 240 °C. It can be seen that the grain size achieves great refinement from the backward extrusion to the fourth extension, whose grain size is 5.2 μm. To be specific, the grain size is reduced by 88.8% compared with the original 46.3 μm. Meanwhile, it can be noticed that a certain number of twins appear during the backward extension and first extension (shown in Figure 6a,b), which are important in the deformation of magnesium alloys. As for the Mg alloys with the structure of HCP, they are difficult to deform because of the short slip system at a low temperature like 240 °C. The twinning does not act on the plastic deformation directly, but adjusts the orientation of the crystal and

release stress concentration, which further stimulates the slipping system and make the slipping and twinning act on the deformation together [26]. With the accumulation of strain, the orientation of the grains turns in the direction that is most favorable to the slip and the twins disappear (shown in Figure 6c,d). Furthermore, the distortion energy stored by twinning and the activation of non-basal slip are responsible for the nucleation of dynamic recrystallization (DRX) [27]. In addition, Figure 6a,b reveal that the "necklace" structure is full of matrix, indicating that DRX occurs and the grain begins to refine [28]. Because of the existence of twins, we can split the coarse grains and promote the nucleation of DRX. At passes 3 and 4, a reduction of the coarse grains can be observed, while there are still a small number of coarse grains due to the low temperature, which cannot provide enough energy to complete it.

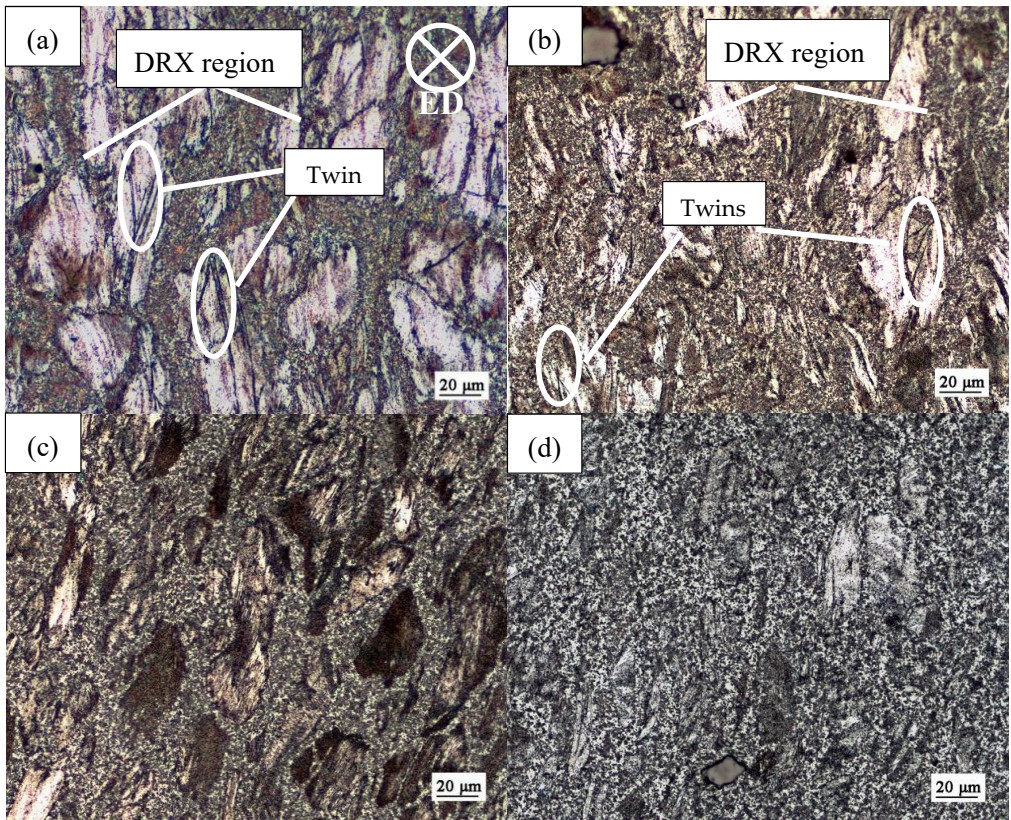

**Figure 6.** Microstructure of different passes at the temperature of 240 °C with different strain percentage: (**a**) backward extrusion, (**b**) extension with 43.8% area reduction, (**c**) extension with 64.0% area reduction, (**d**) extension with 91.0% area reduction.

Scanning electron microscopy (SEM) is often used to characterize the morphology, distribution, and average size of phases [29]. Figure 7 reveals the change of morphology and distribution of second phases after the deformation of 240 °C with different passes. It can be seen that the discontinuous strip-shaped and net-shaped $\beta$-$Mg_{17}Al_{12}$ phases growing from boundaries to grains and needle-shaped $Al_4Ce$ phases with high melting are distributed irregularly at the boundaries, as shown in Figure 7a,b. Meanwhile, the bulk-shaped $Al_8Mn_5$ phases break into small pieces and distribute in the grains. The BSE-SEM shown in Figure 7a (at the right corner) exhibits the contrast of phases: the grain boundaries can be easily separated, grain refinement by twinning can be observed, and the second phases occupy the twins gradually. With the area reduction percentage increased to 43.8% (shown in Figure 7c), it can be noticed that the morphology of $\beta$-$Mg_{17}Al_{12}$ phases is of two different kinds, strip-shaped and granular-shaped, and the distribution of the other second phases gradually becomes uniform. At the same time, the number of twins decreases with the transformation of grain orientation as BSE-SEM,

shown in Figure 7c. Twinning is replaced by slipping and occupies the dominant area of deformation. After the fourth pass of extension, as shown in Figure 7d, the second phases are full of matrix due to the grain refinement and low temperature, which are favorable conditions for second phases to precipitate.

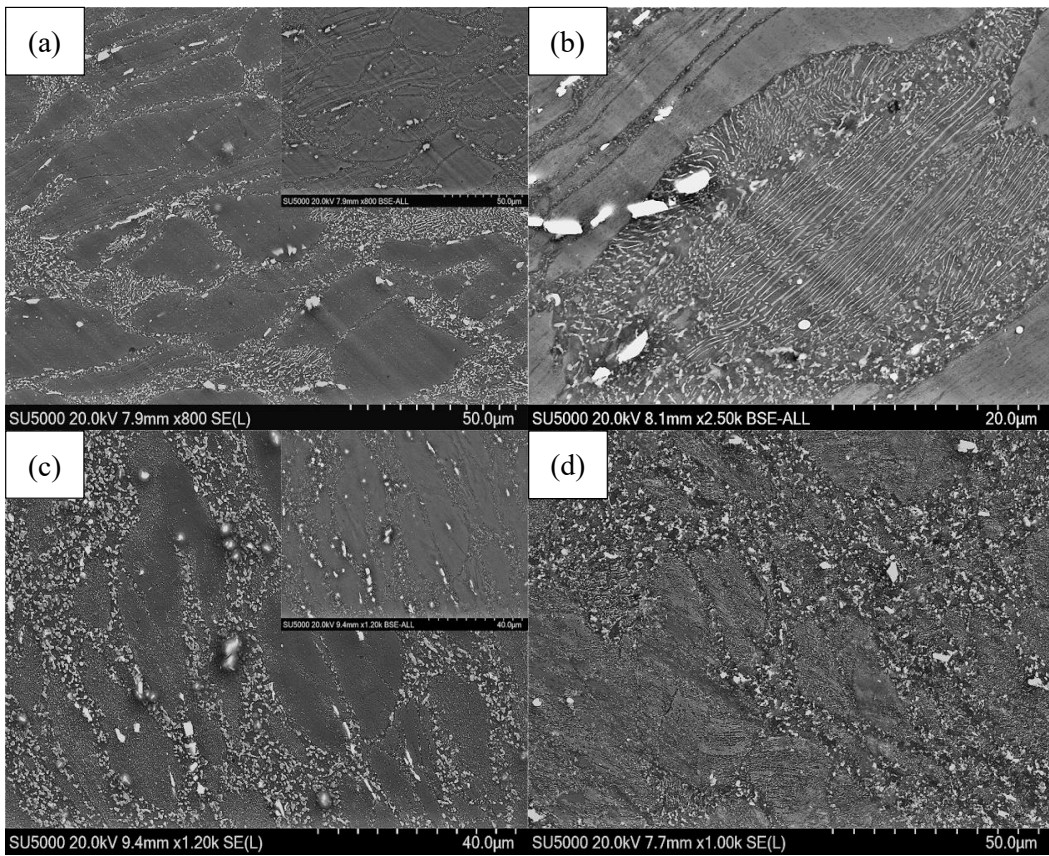

**Figure 7.** SEM images of AZ80 + 0.4%Ce magnesium alloys after deformation at 240 °C with different passes: (**a**) 0 pass (extrusion), (**b**) local enlarged drawing of white region in (**a**), (**c**) first pass (extension), (**d**) fourth pass (extension).

### 3.1.3. Microstructure Evolution at the Same Pass with Different Temperatures

Figure 8 depicts the OM images of magnesium alloys after the first pass with different temperatures. The volume fraction of the second phases undergoes a significant reduction with the increase in temperature. However, $Al_4Ce$ phases, whose formation is due to the addition of Ce with high melting, still occupy the boundaries (as shown in Figure 8d). The rise in temperature increases the solid solubility of Al in Mg, so a large number of Mg-Al eutectic compounds dissolve in the matrix to form a supersaturated solid solution [30]. Meanwhile, the average grain size increases with the increase in temperature. The twins that formed at 270 °C carve up the coarse grains and disappear gradually as the temperature rises (shown in Figure 8a). High temperature not only changes the slip mode from twinning to cross-slipping, but also provides the energy needed for grain growth. In addition, the necklace structure can be seen at all temperatures, which shows that DRX is occurring. At 270 °C and 310 °C, a certain number of coarse grains and fine grains can be seen; the grains grow gradually with the increase in temperature. When the temperature reaches 390 °C, the grains grow so large that they are harmful to the mechanical properties. Thus, 350 °C should be an appropriate deformation temperature.

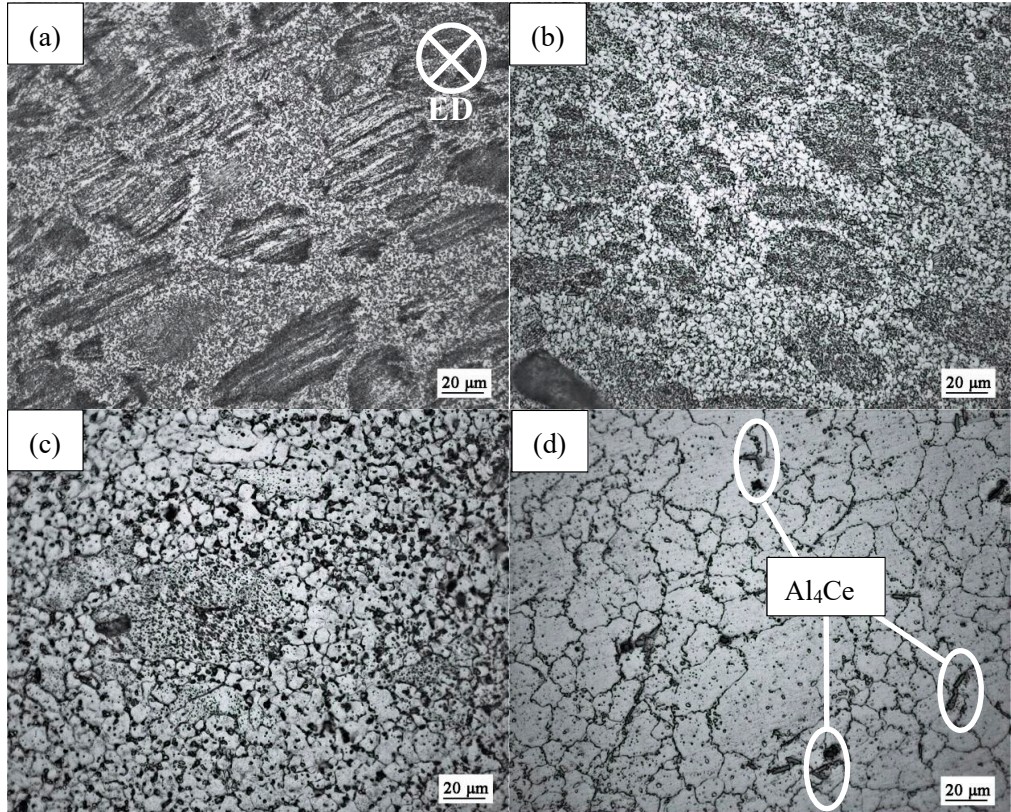

**Figure 8.** OM images of AZ80 + 0.4%Ce magnesium alloys after first pass extension with different temperatures: (**a**) 270 °C, (**b**) 310 °C, (**c**) 350 °C, (**d**) 390 °C.

The OM images of magnesium alloy tubes after fourth pass extension with different temperatures are revealed in Figure 9. The average grain size has been greatly refined and reaches 8.9 μm at 350 °C. Figure 9a reveals that coarse grains still exist and bulk-shaped $Al_8Mn_5$ phases in the grains are not broken into pieces. Compared with 270 °C, the second phases are uniformly distributed in the grain boundaries, while a few coarse grains can be seen in Figure 9b at 310 °C. With the temperature increasing to 350 °C and 370 °C, β-$Mg_{17}Al_{12}$ phases dissolve into the matrix gradually and other phases like $Al_8Mn_5$ and $Al_4Ce$ disperse. However, excessive grain growth occurs during the deformation at 390 °C, which is harmful to the mechanical properties according to the Hall-Petch criterion [31].

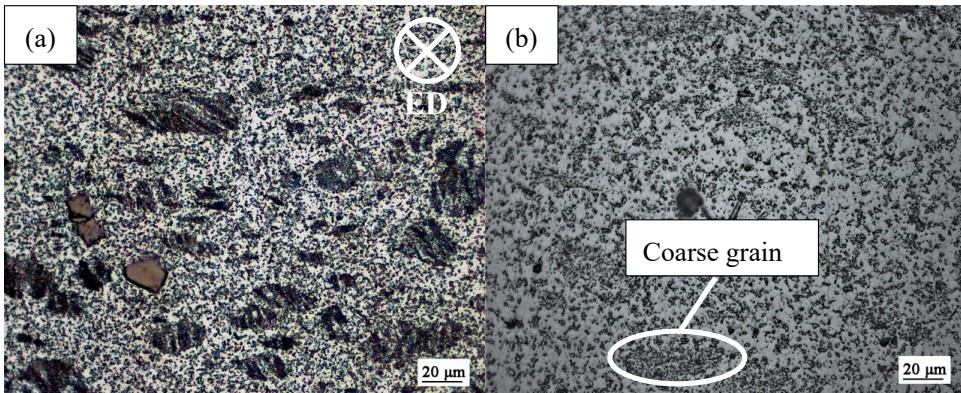

**Figure 9.** *Cont.*

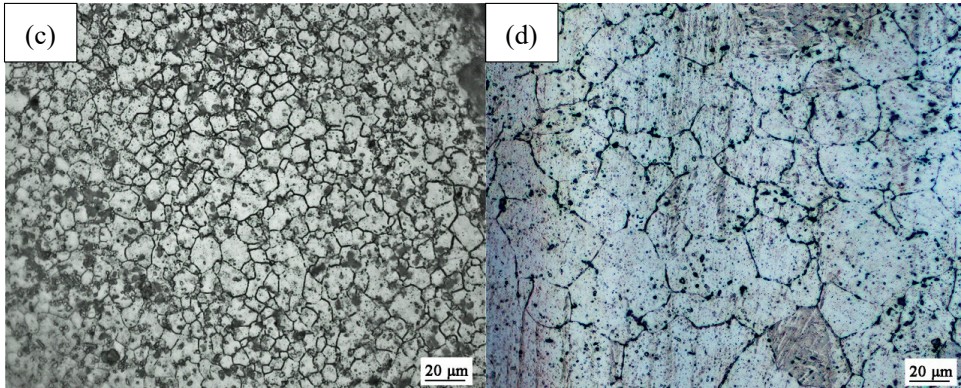

**Figure 9.** OM images of AZ80 + 0.4%Ce magnesium alloys after fourth pass extension with different temperatures: (**a**) 270 °C, (**b**) 310 °C, (**c**) 350 °C, (**d**) 390 °C. SEM images of magnesium alloy tubes after the fourth pass with different temperatures are depicted in Figure 10. It can be noticed that the volume fraction of second phases decreases with the increase in temperature. To be specific, the increase in temperature increases the solid solubility of Al in Mg, a large number of β-$Mg_{17}Al_{12}$ phases dissolve in Mg matrix, and insoluble second phases remain in the boundaries. Therefore, the migration of grain boundaries is greatly improved at a high temperature, which alters the properties of the material. Then, as shown in Figure 10a, the discontinuous net-shaped and strip-shaped β-$Mg_{17}Al_{12}$ phases around the coarse grains grow from the grain boundary to the intragranular region, while the granular-shaped β-$Mg_{17}Al_{12}$ phases around the new grains formed by DRX gather in the grain boundaries. With the breaking up of coarse grains, which are swallowed by DRX grains, the net-shaped β-$Mg_{17}Al_{12}$ phases are gradually replaced by granular β-$Mg_{17}Al_{12}$ phases. Meanwhile, in Figure 10b–d, it can be seen that the $Al_8Mn_5$ phases break into pieces with the increase in area reduction percentage and needle-shaped $Al_4Ce$ phases distributing irregularly at the boundaries. A certain volume fraction of second phases can hinder the migration and growth of grain boundaries.

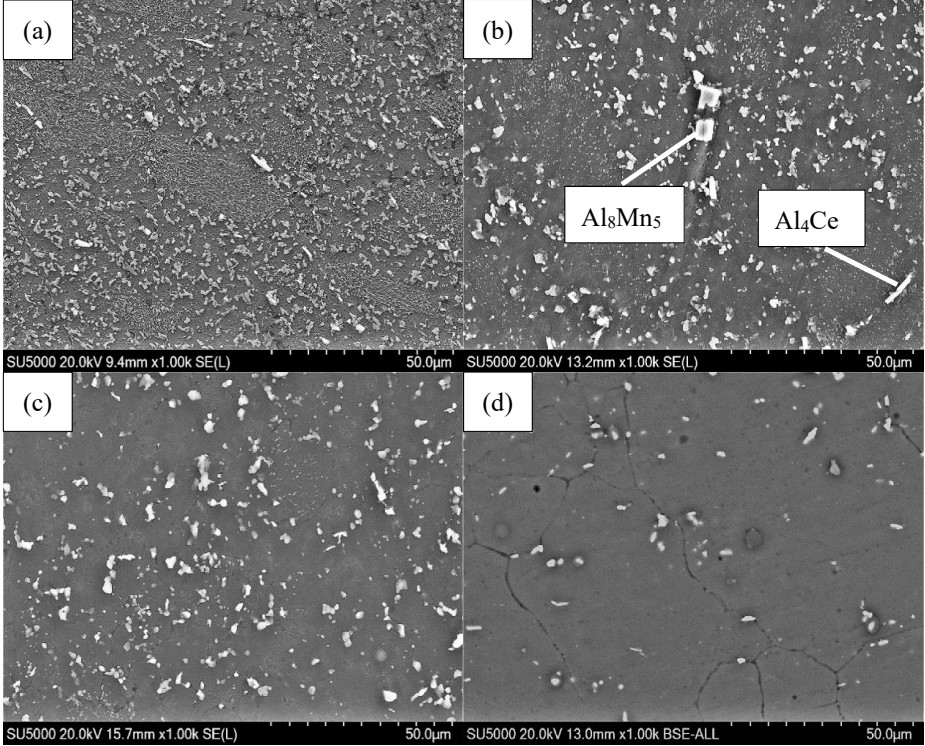

**Figure 10.** SEM images of AZ80 + 0.4%Ce magnesium alloy tubes after fourth pass with different temperatures: (**a**) 270 °C, (**b**) 310 °C, (**c**) 350 °C, (**d**) 390 °C.

### 3.2. Mechanical Properties

Figure 11 shows the Vickers hardness of magnesium alloy tubes without annealing at different passes with different temperatures. It can be seen that the hardness of original $\Phi 10$ mm $\times$ 10 mm pre-extruded specimens is 86 HV. Generally speaking, the hardness grows with the decrease in temperature when it is at the same deformation condition. However, there are also some special points like 270 °C where there is backward extrusion. Second phase precipitation, strengthening by hindering the immigration of grain boundaries and accumulating dislocations, can improve the hardness when the temperature is below 310 °C. The higher the temperature, the faster the precipitation rate. Therefore, the hardness at 310 °C is higher than at 270 °C after backward extrusion. Work hardening greatly improves the hardness with the increase of strain at the same temperature. However, it is not suitable for 240 °C and 270 °C, which involve twinning deformation to coordinate deformation. The line of 240 °C and 270 °C shows that the hardness increases from backwardness extrusion to first pass extension, but the slope of 270 °C increases higher than that of 240 °C, which can be explained by the large number of twins at 240 °C that can dissever coarse grains and lead to twinning dynamic recrystallization (TDRX) softening. With the deformation increase from the first pass to the second pass, the line of 240 °C and 270 °C significantly decreases due to TDRX softening. After that, twins disappeared with the accumulation of strain, and work hardening strengthened the matrix. Therefore, the two lines increase directly. When the temperature reaches 310 °C, the hardness barely changes from backward extrusion to first pass extension, and then the hardness is controlled by work hardening and DRX softening. Finally, the slight increase in hardness should be considered a result of the precipitation of the second phases. The red line shows the change of hardness at 350 °C, and the influence of the second phases disappeared because of the high solid solubility of Al in Mg. A supersaturated solid solution was formed and lattice distortion increased the energy of crystals, which showed the rise in hardness in macroscopic performance to 99 HV. Finally, DRX softening decreased the hardness to 83 HV. At 390 °C, work hardening happened firstly from backward extrusion until the second extension. Then, dynamic recovery occurred and the hardness was further improved until DRX. When the DRX softening is stronger than the work hardening, the hardness begins to decrease, as shown in the line of 390 °C, which shows DRX steady state. The results show that the hardness (84 HV) after fourth pass extension at 350 °C is a little lower than the original specimen (86 HV), which is the reason for non-annealing.

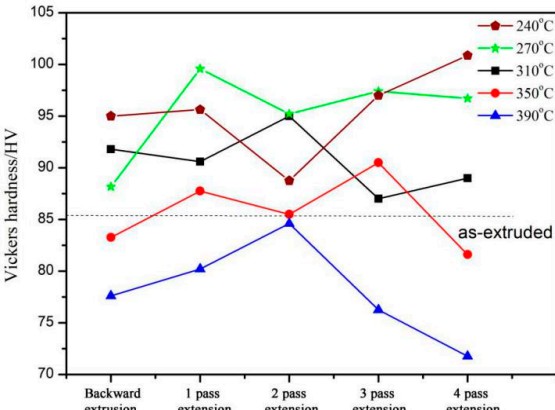

**Figure 11.** Vickers hardness of magnesium alloy tubes at different passes with different temperatures without annealing.

## 4. Conclusions

A new method and molds with hot deformation to replace the cold drawing process was proposed to fabricate magnesium ultra-thin-walled tubes. The die structure and experimental method were

given, and AZ80 + 0.4%Ce magnesium alloys were used to carry out the experiment for preliminary verification. The following conclusions can be drawn:

(1) The molds were designed and thin-walled tubes of AZ80 + 0.4%Ce magnesium alloy with an inside diameter of 6.0 mm and a wall thickness of 0.6 mm were fabricated by hot multi-pass variable wall thickness back extrusion, which can be split into two processes: hot backward extrusion and four-pass hot extension with variable diameters. The success of AZ80 + 0.4%Ce can provide a reference for other magnesium alloys to fabricate ultra-thin walled tubes for electronic components and medical devices.

(2) According to the microstructure analysis, the grain size decreased from 46.3 μm to 8.9 μm with four-pass extension at a deformation temperature of 350 °C and a punch speed of 0.1 mm/s, while there was excessive grain growth at 390 °C and non-uniformity of grain sizes and phases at 310 °C.

(3) Twinning occurred when the deformation temperature was below 310 °C, and disappeared gradually with the increase in accumulated strain. Twinning, precipitation of second phases, twinning dynamic recrystallization (TDRX), and work hardening were combined to change the hardness of tubes at 240 °C and 270 °C. The formation of twins hindered the slip of dislocations and refined grains at the same time. When the temperature reaches 350 °C and higher, the competitive mechanism between work hardening and DRX softening dominates plastic deformation. The hardness reached 93HV after the third extension without annealing at 350 °C and would be higher after heat treatment.

## 5. Patents

In order to protect the intellectual property rights, the multi-pass variable wall thickness extrusion molds have been patented and authorized (ZL 2017 1 0548593.9, China). This method of fabrication of ultra-thin walled tubes has been patented and authorized (ZL 2017 1 0548594.3, China).

**Author Contributions:** Z.Z. and M.F. designed the experiments; Z.Y., Z.L., and G.Z. conducted the experiments and collected the data; J.Z. and Y.W. analyzed the data; Z.Y. wrote the paper.

**Funding:** The present study was supported by the National Natural Science Foundation of China (NSFC) under grant No. 51775520 and No. 51675492, and the Natural Science Foundation of Shanxi Province under grant No. 201801D121106.

**Conflicts of Interest:** The authors declare no conflict of interest.

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
