# Peer review of "Research on AZ80 + 0.4%Ce (wt %) Ultra-Thin-Walled Tubes of Magnesium Alloys: The Forming Process, Microstructure Evolution and Mechanical Properties"

_metals, doi:10.3390/met9050563_

Round 1

Reviewer 1 Report

The work entitled “Research on Biomedical Ultra-thin-walled Tube of Magnesium Alloys: New forming process, Microstructure Evolution and mechanical properties” describes the fabrication of AZ80+0.4%Ce Mg alloy thin-walled tubes (with 5 passes). Even though the work is interesting there are many details missing (see list below). The mechanical characterization is well done and well discussed but never in light of the actual application. This needs improvement. The English language also needs refinement. There are many mistakes along the manuscript that difficult reading and understanding of the information the authors want to convey. Most importantly, when researching new materials for biomedical applications, cytotoxicity tests must be employed, which are missing from the present work.

Introduction:

The goal and particularly the novelty of the research should be more explicit in the introduction.

Materials and Methods:

Section 2.2: there are no brands or suppliers referred to for any of the materials used in the experiments.

Section 2.3: there are many details missing from the techniques XRD and SEM, please include them.

Results and Discussion:

The results are well presented and discussed. However, discussion considering the actual application of these materials is missing. There is no information considering the final structure of this stents and the advantages they will bring compared to existing alternatives.

Conclusions:

Reflect the results and discussion and fail again to convey the importance of this study for the actual application and the potentialities of these findings.

Author Response

Dear professor;

        Thanks for your suggestions on this article. I have already answered your question as follows.

(1) The original intention of this research is to propose a new forming method to replace traditional cold drawing to fabricate ultra-thin walled tube. AZ80+0.4%Ce Mg alloy  is just a preliminary proof of this idea. Meanwhile, stent is just one of the applications of ultra-thin walled tube. which was overrendered in previous article. Thus, I have revised the content of the article and marked it in red.

(2) I have made it clear to introduce the goal and particularly the novelty of the research in the part of "Introduction" which was marked in red.

(3) I have pointed out the brands of materials used in the experiments in section 2.2, and gave more details of SEM and XRD in section 2.3 which were also marked in red.

(4) The patents of moulds and method to fabricate the ultra-thin walled tube have been authorized, and the primary verification was sucessful by using AZ80+0.4%Ce. It will soon be applied to other metals. 

Best wishes.

Zhaoming Yan

Reviewer 2 Report

1-      The authors need to check the manuscript for the typo errors. For instance, in lines 65 and 76 (clod drawing) need to be changed to (cold drawing).

2-      Although the stent fabrication is very important for biomedical application, AZ Mg alloys are not suitable for biomedical applications as the presence of Al can cause Alzheimer's disease/Dementia. 

3-      Though extensive study carried out to understand the microstructure of the Mg ultra-thin-walled tubes after deformation, the electrochemical study is important to understand the effect of deformation on electrochemical behaviour during exposure to the corrosive environment.   

Author Response

Dear professor;

       Thanks for your suggetions on this article. I have already answered your questions as follows. I have revised part of "Introduction" where the application in stent was overrendered, which may be mislead the readers. The main purpose of the work is introduceing a new forming technology with large area reduction which the cold drawing can not reach. Meawhile, AZ80+0.4%Ce Mg alloy was used to verify the feasibility and the improvement of microstructure and mechanical properties. The patents of moulds and method to fabricate the ultra-thin walled tubes were authoried and the application in the other metals would be soon.

Point 1: The authors need to check the manuscript for the typo errors. For instance, in lines 65 and 76 (clod drawing) need to be changed to (cold drawing).  

Resonse 1: I have corrected the typo errors in the manuscrit which were marked in red.

Point 2:  Although the stent fabrication is very important for biomedical application, AZ Mg alloys are not suitable for biomedical applications as the presence of Al can cause Alzheimer's disease/Dementia. 

Response 2: The original intension of the work is to propose a new forming method to replace the traditional cold drawing to fabricate ultra-thin walled tube. AZ80+0.4%Ce Mg alloy is just a preliminary proof of the idea. Meamwhile, stent is just one of the applications of ultra-thin walled tube, which was overrendered in previos manuscript. Thus, I have revised the content of article and marked it in red.

Point 3:  Though extensive study carried out to understand the microstructure of the Mg ultra-thin-walled tubes after deformation, the electrochemical study is important to understand the effect of deformation on electrochemical behaviour during exposure to the corrosive environment.  

Response 3: AZ80+0.4%Ce Mg alloy was only used to verify the feasibility and improvement of mechanical properties. Since the stent is not the main discussion in the manuscript, eletrochemical behaviour is not an important indicator.

Please refer to my revised manuscript for seprcific amendments.

Best wishes. 

Round 2

Reviewer 1 Report

The authors responded to all comments and made improvements where recommended.